# No general relationship between mass and temperature in endothermic species

Kristina Riemer[1]*, Robert P Guralnick[2], Ethan P White[1,3]

[1]Department of Wildlife Ecology and Conservation, University of Florida, Gainesville, United States; [2]Department of Natural History, University of Florida, Gainesville, United States; [3]Informatics Institute, University of Florida, Gainesville, United States

**Abstract** Bergmann's rule is a widely-accepted biogeographic rule stating that individuals within a species are smaller in warmer environments. While there are many single-species studies and integrative reviews documenting this pattern, a data-intensive approach has not been used yet to determine the generality of this pattern. We assessed the strength and direction of the intraspecific relationship between temperature and individual mass for 952 bird and mammal species. For eighty-seven percent of species, temperature explained less than 10% of variation in mass, and for 79% of species the correlation was not statistically significant. These results suggest that Bergmann's rule is not general and temperature is not a dominant driver of biogeographic variation in mass. Further understanding of size variation will require integrating multiple processes that influence size. The lack of dominant temperature forcing weakens the justification for the hypothesis that global warming could result in widespread decreases in body size.
DOI: https://doi.org/10.7554/eLife.27166.001

## Introduction

Bergmann's rule describes a negative relationship between body mass and temperature across space that is believed to be common in endothermic species (*Bergmann, 1847*; *Brown and Lee, 1969*; *Kendeigh, 1969*; *Freckleton et al., 2003*; *Carotenuto et al., 2015*). Many hypotheses have been proposed to explain this pattern (*Blackburn et al., 1999*; *Ashton, 2002*; *Watt et al., 2010*) including the heat loss hypothesis, which argues that the higher surface area to volume ratio of smaller individuals results in improved heat dissipation in hot environments (*Bergmann, 1847*). Though originally described for closely-related species (i.e., interspecific; *Blackburn et al., 1999*), the majority of studies have focused on the intraspecific form of Bergmann's rule (*Rensch, 1938*; *Meiri, 2011*) by assessing trends in individual size within a species (*Langvatn and Albon, 1986*; *Yom-Tov and Geffen, 2006*; *Gardner et al., 2009*). Bergmann's rule has been questioned both empirically and mechanistically (*McNab, 1971*; *Geist, 1987*; *Huston and Wolverton, 2011*; *Teplitsky and Millien, 2014*) but the common consensus from recent reviews is that the pattern is general (*Ashton et al., 2000*; *Ashton, 2002*; *Meiri and Dayan, 2003*; *Watt and Salewski, 2011*).

It has recently been suggested that this negative relationship between mass and temperature could result in decreasing individual size across species in response to climate change (*Sheridan and Bickford, 2011*) and that this may be a 'third universal response to warming' (*Gardner et al., 2011*). The resulting shifts in size distributions could significantly alter ecological communities (*Brose et al., 2012*), especially if the rate of size decrease varies among species (*Sheridan and Bickford, 2011*). While there is limited empirical research on body size responses to changes in temperature through time (but see *Smith et al., 1995*; *Caruso et al., 2014*; *Teplitsky and Millien, 2014*), the apparent generality of Bergmann's rule across space indicates the likelihood of a similar relationship in response to temperature change across time.

*For correspondence:
kristina.riemer@weecology.org

Competing interests: The authors declare that no competing interests exist.

**eLife digest**  Scientists have found that individual animals of the same species tend to be smaller in hotter environments and larger in cooler ones. They named this pattern "Bergmann's Rule" to describe how temperature can influence the size of an animal. However, most studies of Bergmann's Rule have only looked at one or a few species at a time.

Knowing how many species follow this rule is important because globally rising temperatures could cause lots of species to become smaller. Since the size of organisms affects how much food and space they need, this could disrupt natural systems around the world.

To test if Bergmann's rule can be extended to many species, Riemer, Guralnick, and White assessed the relationship between temperature and body mass for 952 bird and mammal species. Contrary to Bergmann's Rule, the results showed that most of the species had similar sizes regardless of the temperature of their environment. Only about 140 species were smaller in hotter environments, and about 70 species were larger in hotter environments. This suggest that Bergmann's Rule does not apply to most species as expected.

While most birds and mammals may not get bigger or smaller due to warming global temperatures, the few species that do change in size – and the species that interact with them – may be more likely to become endangered or extinct. If we can determine which animals are at risk, we can prioritize their conservation and design better plans for doing so. Losing even a single species disrupts our ecosystems, on which we rely for critical resources like food, building materials, and clean air.

DOI: https://doi.org/10.7554/eLife.27166.002

The generality of Bergmann's rule is based on many individual studies that analyze empirical data on body size across an environmental gradient (e.g., *Langvatn and Albon, 1986*; *Barnett, 1977*; *Fuentes and Jaksic, 1979*; *Dayan et al., 1989*; *Sand et al., 1995*) and reviews that compile and evaluate the results from these studies (*Ashton, 2002*; *Meiri and Dayan, 2003*; *Watt et al., 2010*). Most individual studies of Bergmann's rule are limited by: (1) analyzing only one or a few species (e.g., *Langvatn and Albon, 1986*); (2) using small numbers of observations (e.g., *Fuentes and Jaksic, 1979*); (3) only including data at the small scales typical of ecological studies (e.g., *Sand et al., 1995*); (4) using latitude instead of directly assessing temperature (e.g., *Barnett, 1977*); and (5) focusing on statistical significance instead of the strength of the relationship (e.g., *Dayan et al., 1989*). The reviews tabulate the results of these individual studies and assess patterns in the direction and significance of relationships across species. Such aggregation of published results allows for a more general understanding of the pattern but, in addition to limitations of the underlying studies, the conclusions may be influenced by publication bias and selective reporting due to studies or individual analyses that do not support Bergmann's rule being published less frequently (*Koricheva et al., 2013*).

Previous analyses of publication bias in the context of Bergmann's rule have found no evidence for selective publication, which supports the idea that it is a general rule (*Ashton, 2002*; *Meiri et al., 2004*). However, two of the most extensive studies of Bergmann's rule, which both used museum records to assess dozens of intraspecific Bergmann's rule relationships simultaneously, found that the majority of species did not exhibit significant positive relationships between latitude and size (*McNab, 1971*; *Meiri et al., 2004*). As a result, understanding the generality of this ecophysiological rule and its potential implications for global change requires more extensive analysis.

A data-intensive approach to analyzing Bergmann's rule, evaluating the pattern using large amounts of broad scale data, has the potential to overcome existing limitations in the literature and provide a new perspective on the generality of the intraspecific form of Bergmann's rule. Understanding the generality of the temperature-mass relationship has important implications for how size will respond to climate change. We use data from VertNet (*Constable et al., 2010*), a large compilation of digitized museum records that contains over 700,000 globally distributed individual-level size measures, to evaluate the intraspecific relationship between temperature and mass for 952 mammal and bird species. The usable data consist of 273,901 individuals with an average of 288 individuals per species, with individuals of each species spanning an average of 75 years and 34 latitudinal

degrees. This approach reduces or removes many of the limitations to previous approaches and the results suggest that Bergmann's rule is not a strong or general pattern.

## Results

Most of the species in this study showed weak non-significant relationships between temperature and mass (*Figures 1* and *2*). The distribution of correlation coefficients was centered near zero with a mean correlation coefficient of −0.05 across species (*Figure 2A*). Relationships for 79% of species were not significantly different from zero based on false discovery rate-controlled p values and associated z scores, while 14% of species' relationships were significant and negative and 7% were significant and positive (*Figure 2A*, *Figure 2—figure supplement 1*). Temperature explained less than 10% of variation in mass (i.e., $-0.316 < r < 0.316$) for 87% of species, indicating that temperature explained very little of the observed variation in mass for these species (*Figure 2A*).

The weak, non-directional intraspecific relationships indicated by the distribution of correlation coefficients are consistent across taxonomic groups and temporal lags. Mean correlation coefficients for both endothermic classes are −0.006 and −0.065, for mammals and birds respectively (*Figure 2B*). Similarly, correlation coefficient distributions were approximately centered on zero for all of the 30 orders analyzed ($-0.2 < \bar{r} < 0.003$ for orders with more than 10 species; *Figure 3* and *Figure 3—figure supplement 1*), and for migrant and nonmigrant bird species (*Figure 2—figure supplement 2*). Correlation coefficient distributions for temperature-mass relationships using lagged temperatures were centered around zero like those using temperature from the collection year (*Figure 4* and *Figure 4—figure supplement 1*), indicating that there was not a temporal lag effect on the response of species' masses to temperature. Correlation coefficients did not vary systematically by sample size (*Figure 5A*), extent of variation in temperature or mass (*Figure 5B,C*), species' average mass (*Figure 5D*), or species' average latitude (*Figure 5E*). While temperature is considered the actual driver, some studies use latitude as a proxy when evaluating variation in size (*Bergmann, 1847*; *Stillwell, 2010*). Using latitude, the mean correlation coefficient was −0.05 with no statistically significant latitude-mass relationship for 71% of species (*Figure 2—figure supplement 3*), while the respective values for temperature were −0.05 and 79% (*Figure 2A*). Results were robust to a variety of decisions and stringencies about how to filter the size (*Figure 2—figure supplements 4* and *5*) and species data (*Figure 2—figure supplements 6* and *7*).

## Discussion

In contrast to conventional wisdom and several recent review papers, our analysis of 952 species shows little to no support for a negative intraspecific temperature-mass relationship that is sufficiently strong or common to be considered a biogeographic rule. Three quarters of bird and mammal species show no significant change in mass across a temperature gradient and temperature explained less than 10% of intraspecific variation in mass for 87% of species (*Figure 2A*). This was true regardless of taxonomic group (*Figures 2* and *3*), temporal lag in temperature (*Figure 4*), species' size, location, or sampling intensity or extent (*Figure 5*). These results are consistent with two previous studies that examined museum specimen size measurements across latitude. The first study showed that 22 out of 47 North American mammal species studied had no relationship between latitude and length, and 10 of the 25 significant relationships were opposite the expected direction (*McNab, 1971*). The second found a similar proportion of non-significant results (42/87), but a lower proportion of significant relationships that opposed the rule (9/45) for carnivorous mammals (*Meiri et al., 2004*). While more species had significant negative relationships than positive in both our study and these two museum-based studies, in all cases less than half of species had significant negative correlations (14–41%). In combination with these two smaller studies, our results suggest that there is little evidence for a strong or general Bergmann's rule when analyzing raw data instead of summarizing published results.

Our results are inconsistent with recent reviews, which have reported that the majority of species conform to Bergmann's rule (*Ashton, 2002*; *Meiri and Dayan, 2003*; *Watt et al., 2010*). While these reviews included results that were either non-significant or opposite of Bergmann's rule, the proportion of significant results in support of Bergmann's rule was higher and therefore resulted in conclusions that supported the generality of the temperature-mass relationship. Generalizing from results

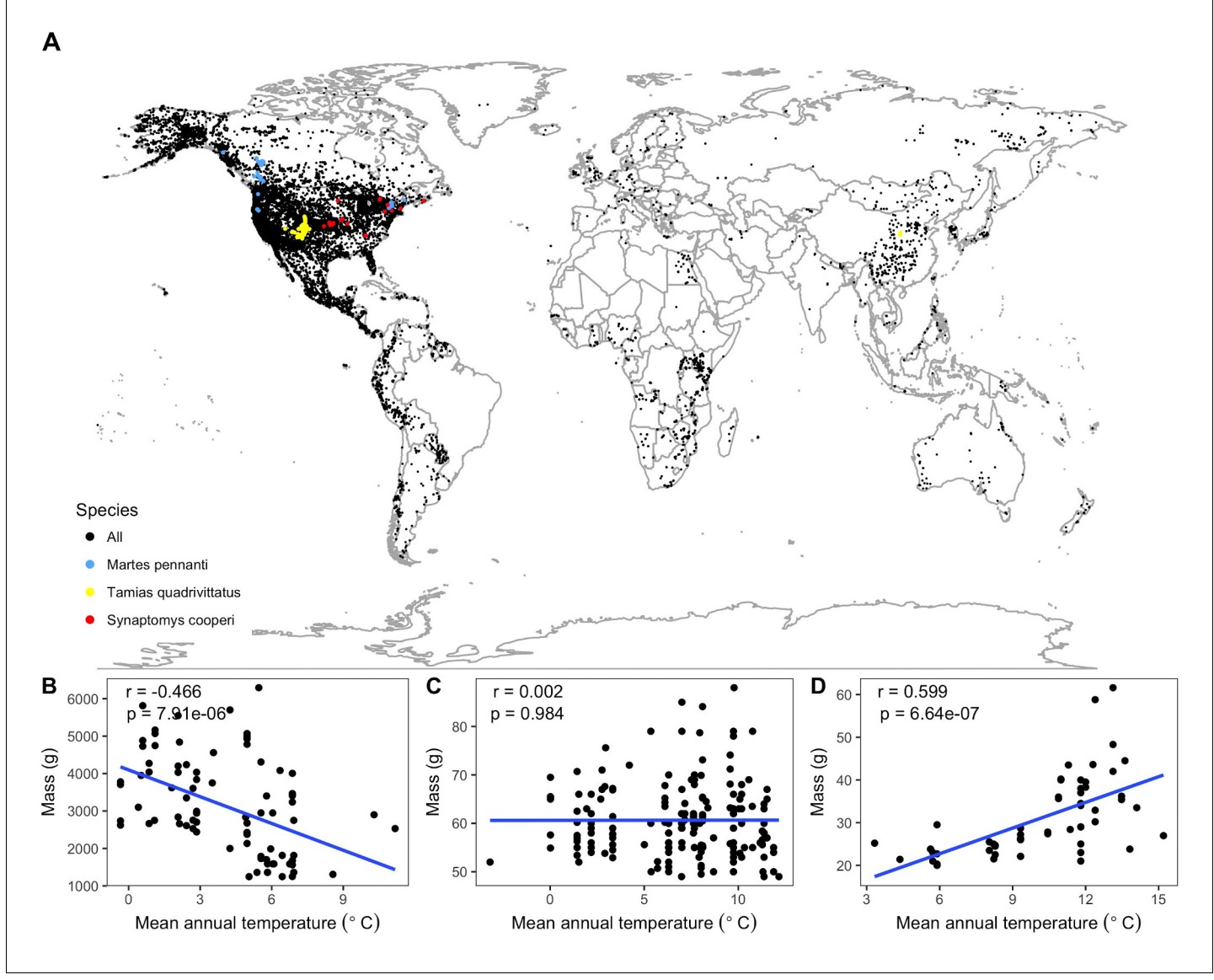

**Figure 1.** Species spatial distributions and selected temperature-mass relationships. (A) Spatial collection locations of all individual specimens. All species shown with black points except three species, whose relationships between mean annual temperature and mass are shown at bottom (B–D), are marked with colored points. These species were chosen to represent the range of variability in relationship strength and direction exhibited by the 952 species from the study: *Martes pennanti* had a negative relationship with temperature explaining a substantial amount of variation in mass (B; blue points); *Tamias quadrivittatus* had no directional relationship between temperature and mass with temperature having little explanatory power (C; yellow points); *Synaptomys cooperi* had a strong positive temperature-mass relationship with a correlation coefficient (r) in the 99[th] percentile of all species' values (D; red points). Intraspecific temperature-mass relationships are shown with black circles for all individuals and ordinary least squares regression trends as blue lines. Linear regression correlation coefficients and p-values in upper left hand corner of figure for each species. For remaining species relationships, see *Figure 1—figure supplement 1–12*.

DOI: https://doi.org/10.7554/eLife.27166.003

The following figure supplements are available for figure 1:

**Figure supplement 1.** Species' temperature-mass relationships.
DOI: https://doi.org/10.7554/eLife.27166.004
**Figure supplement 2.** Species' temperature-mass relationships.
DOI: https://doi.org/10.7554/eLife.27166.005
**Figure supplement 3.** Species' temperature-mass relationships.
DOI: https://doi.org/10.7554/eLife.27166.006
**Figure supplement 4.** Species' temperature-mass relationships.

*Figure 1 continued*

DOI: https://doi.org/10.7554/eLife.27166.007

**Figure supplement 5.** Species' temperature-mass relationships.
DOI: https://doi.org/10.7554/eLife.27166.008
**Figure supplement 6.** Species' temperature-mass relationships.
DOI: https://doi.org/10.7554/eLife.27166.009
**Figure supplement 7.** Species' temperature-mass relationships.
DOI: https://doi.org/10.7554/eLife.27166.010
**Figure supplement 8.** Species' temperature-mass relationships.
DOI: https://doi.org/10.7554/eLife.27166.011
**Figure supplement 9.** Species' temperature-mass relationships.
DOI: https://doi.org/10.7554/eLife.27166.012
**Figure supplement 10.** Species' temperature-mass relationships.
DOI: https://doi.org/10.7554/eLife.27166.013
**Figure supplement 11.** Species' temperature-mass relationships.
DOI: https://doi.org/10.7554/eLife.27166.014
**Figure supplement 12.** Species' temperature-mass relationships.
DOI: https://doi.org/10.7554/eLife.27166.015

in the published literature involves the common challenges of publication bias and selective reporting (*Koricheva et al., 2013*). In addition, because the underlying Bergmann's rule studies typically report minimal statistical information, often providing only relationship significance or direction instead of p-values or correlation coefficients (*Meiri and Dayan, 2003*), proper meta-analyses and associated assessments of biological significance are not possible. While several reviews found no evidence for publication bias using limited analyses (*Ashton, 2002*; *Meiri et al., 2004*), the notable differences between the conclusions of our data-intensive approach and those from reviews suggest that publication bias in literature examining Bergmann's rule warrants further investigation. These differences also demonstrate the value of data-intensive approaches in ecology for overcoming potential weaknesses and biases in the published literature. Directly analyzing large quantities of data from hundreds of species allows us to assess the generality of patterns originally reported in smaller studies while avoiding the risk of publication bias. This approach additionally makes it easier to integrate other factors that potentially influence size into future analyses. The new insight gained from this data-intensive approach demonstrates the value of investing in large compilations of ecologically-relevant data (*Hampton et al., 2013*) and the proper training required to work with these datasets (*Hampton et al., 2017*).

Our analyses and conclusions are limited to the intraspecific form of Bergmann's rule. This is the most commonly studied and well-defined form of the relationship, and the one most amenable to analyses using large compilations of museum data. Difficulty in interpreting Bergmann's original formulation has resulted in an array of different ideas and implementations of interspecific analyses (*Blackburn et al., 1999*; *Meiri and Thomas, 2007*; *Watt et al., 2010*; *Meiri, 2011*). The most common forms of these interspecific analyses involve correlations between various species-level size metrics and environmental measures and are conducted at various taxonomic levels from genus to class (e.g., *Blackburn and Gaston, 1996*; *Diniz-Filho et al., 2007*; *Boyer et al., 2009*; *Clauss et al., 2013*). Efforts to apply data-intensive approaches to the interspecific form of this relationship will need to address the fact that occurrence records are not evenly distributed across the geographic range of species, and determine how the many interpretations of interspecific Bergmann's rule are related to one another and the biological expectations for interspecific responses to temperature.

The original formulation of Bergmann's rule, and the scope of our conclusions, apply only to endotherms. However, negative temperature-mass relationships have also been documented in ectotherms, with the pattern referred to as the size-temperature rule (*Ray, 1960*; *Angilletta and Dunham, 2003*). In contrast to the hypotheses for Bergmann's rule, which are based primarily on homeostasis (*Gardner et al., 2011*), the size-temperature rule in ectotherms is thought to result from differences between growth and development rates (*Forster et al., 2011*). The current version of VertNet contained ectotherm size data for only seven species, which is not sufficient to complete a comprehensive analysis of the ectotherm size-temperature rule. Future work exploring the

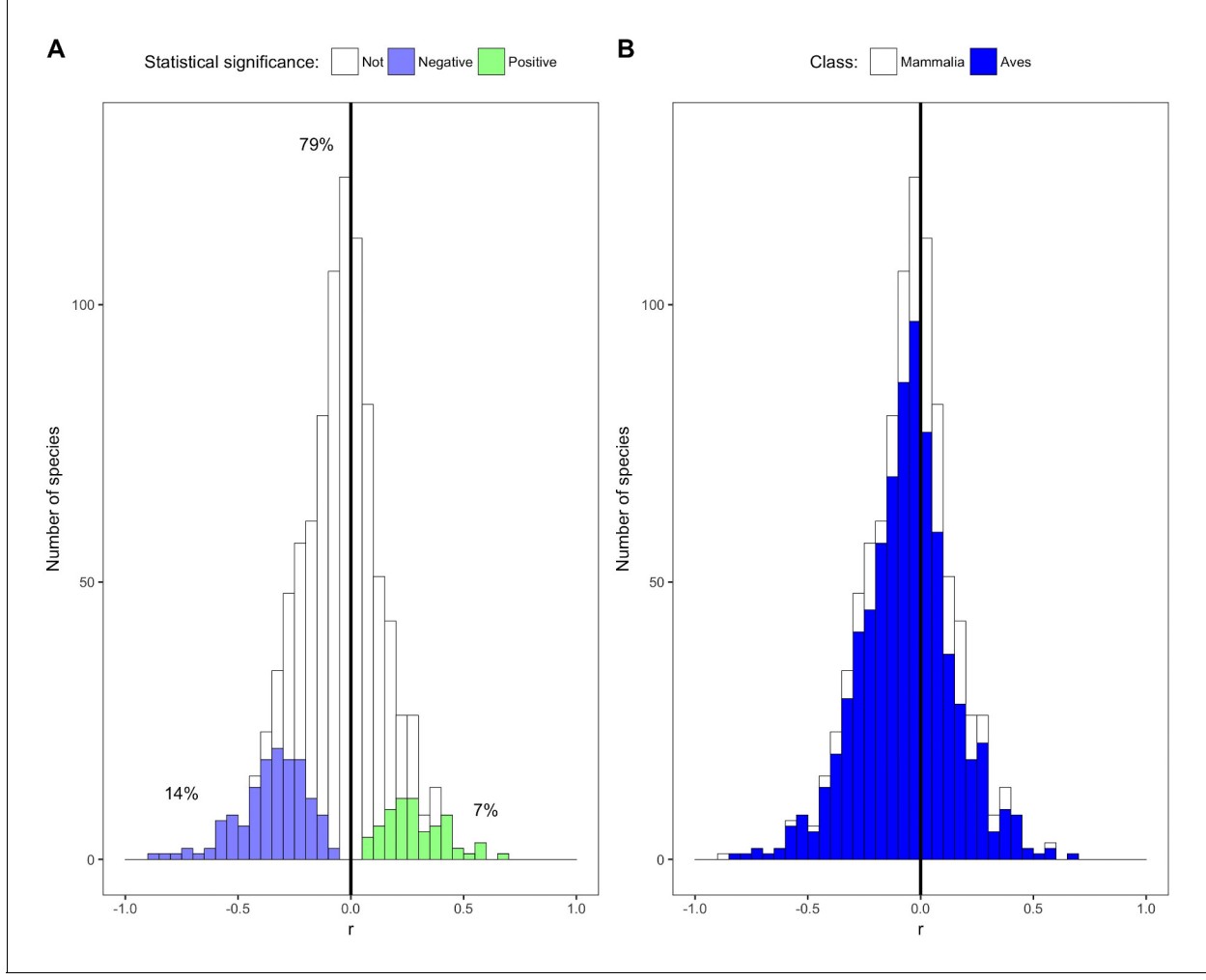

**Figure 2.** Species correlation coefficients by statistical significance and taxonomic class. (A) Stacked histogram of correlation coefficients (r) for all species' intraspecific temperature-mass relationships. Colored bars show species with statistically significant relationships, both negative (purple) and positive (green), while white bars indicate species with relationship slopes that are not significantly different from zero. Percentages are of species in each group. (B) Stacked histogram of all species' correlation coefficients with bar color corresponding to taxonomic class. Dark vertical lines are correlation coefficients of zero. See *Figure 2—figure supplements 1–6*.

DOI: https://doi.org/10.7554/eLife.27166.016

The following figure supplements are available for figure 2:

**Figure supplement 1.** Species z scores and z distribution.
DOI: https://doi.org/10.7554/eLife.27166.017

**Figure supplement 2.** Species correlation coefficients by bird migratory status.
DOI: https://doi.org/10.7554/eLife.27166.018

**Figure supplement 3.** Species correlation coefficients for latitude-mass relationships.
DOI: https://doi.org/10.7554/eLife.27166.019

**Figure supplement 4.** Species correlation coefficients for temperature-mass relationships with lifestage sensitivity analysis.
DOI: https://doi.org/10.7554/eLife.27166.020

**Figure supplement 5.** Species correlation coefficients for temperature-mass relationships with outlier sensitivity analysis.
DOI: https://doi.org/10.7554/eLife.27166.021

**Figure supplement 6.** Species correlation coefficients for temperature-mass relationships with species thresholds increased.
DOI: https://doi.org/10.7554/eLife.27166.022

**Figure supplement 7.** Species correlation coefficients for temperature-mass relationships with species thresholds decreased.
DOI: https://doi.org/10.7554/eLife.27166.023

ectotherm size-temperature rule in natural systems using data-intensive approaches is necessary for understanding the generality of this rule in ectotherms, and data may be sought for this effort in the literature or via a coordinated effort by museums to continue digitizing size measurements for specimens.

A number of mechanisms have been suggested to explain why higher temperatures should result in lower body sizes, including heat loss, starvation, resource availability, migratory ability, and phylogenetic constraints (*Blackburn et al., 1999*). Most of the proposed hypotheses have not been tested sufficiently to allow for strong conclusions to be drawn about their potential to produce Bergmann's rule (*Blackburn et al., 1999*; *Watt et al., 2010*; *Teplitsky and Millien, 2014*) and the widely studied heat loss hypothesis has been questioned for a variety of reasons (*James, 1970*; *McNab, 1971*; *Blackburn et al., 1999*; *Watt et al., 2010*; *McNamara et al., 2016*). While no existing hypotheses have been confirmed, it is possible that some processes are producing negative relationships between size and temperature. The lack of a strong relationship does not preclude

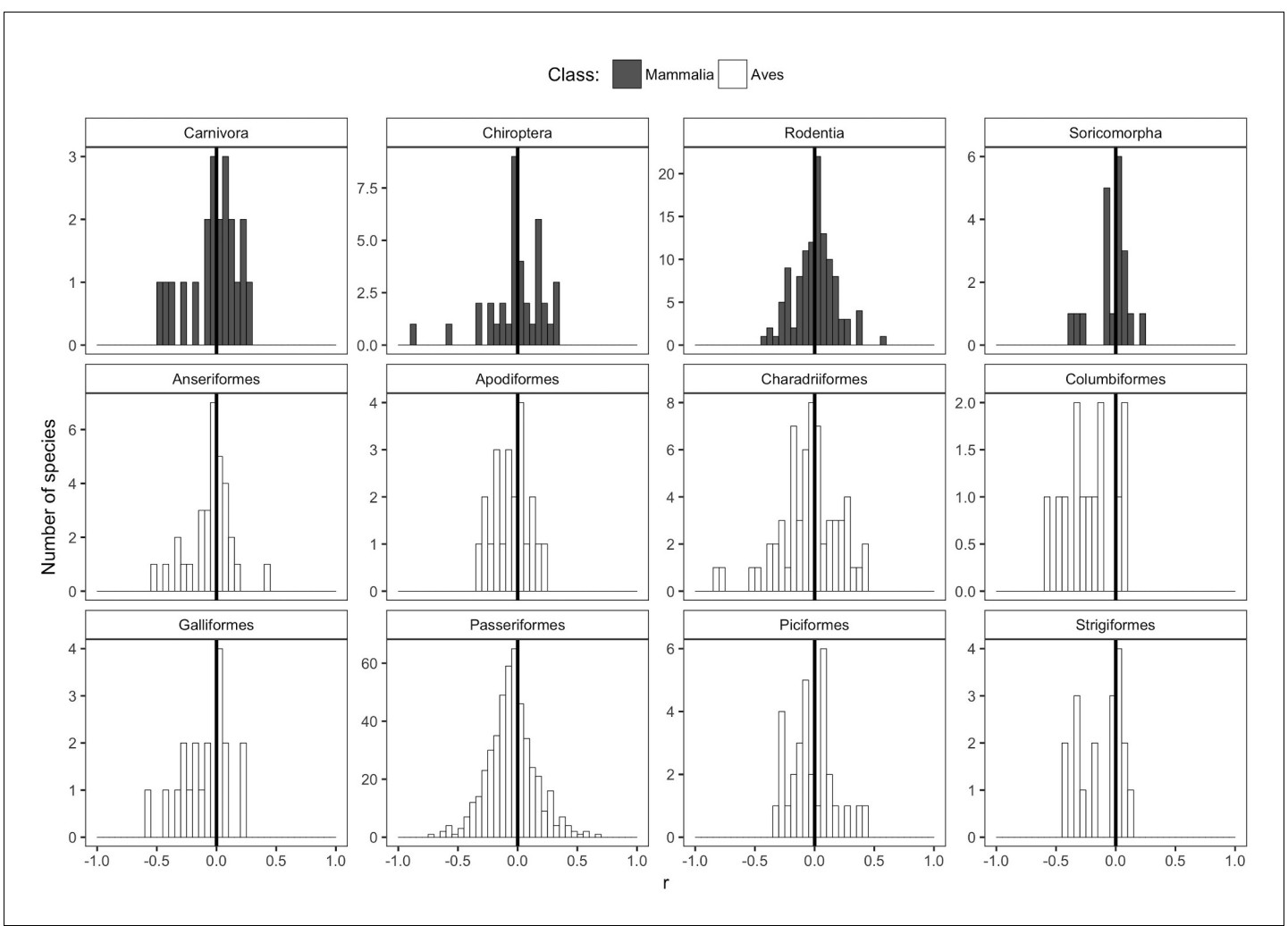

**Figure 3.** Species correlation coefficients for selected taxonomic orders. Histograms of correlation coefficients (r) from intraspecific temperature-mass relationships for each taxonomic order represented by more than ten species, with order shown above histogram. Height of y-axis varies depending on number of species. Bar color indicates taxonomic class. Dark vertical lines are correlation coefficients of zero. For remaining orders, see *Figure 3—figure supplement 1*.

DOI: https://doi.org/10.7554/eLife.27166.024

The following figure supplement is available for figure 3:

**Figure supplement 1.** Species correlation coefficients for remaining taxonomic orders.

DOI: https://doi.org/10.7554/eLife.27166.025

processes that result in a negative temperature-mass relationship, but it does suggest that these processes have less influence relative to other factors that affect intraspecific size.

The relative importance of the many factors besides temperature that can influence size within a species is as yet unknown. Size is affected by abiotic factors such as humidity and resource availability (*Teplitsky and Millien, 2014*), characteristics of individuals like clutch size (*Boyer et al., 2009*), and community context, including possible gaps in size-related niches (*Smith et al., 2010*) and the trophic effects of primary productivity on consumer size (*Sheridan and Bickford, 2011*). Temperature itself can have indirect effects on size, such as via habitat changes in water flow or food availability, that result in size responses opposite of Bergmann's rule (*Gardner et al., 2011*). Anthropogenic influences have been shown to influence the effect of temperature on size (*Faurby and Araújo, 2016*), and similar impacts of dispersal, extinctions, and the varying scales of climate change have been proposed (*Clauss et al., 2013*). Because our data primarily came from North America, further analyses focused on species native to other continents could reveal differing temperature-mass relationships due to varying temperature regimes. While our work shows that more species have negative significant relationships between temperature and mass than positive, only 21% of species have statistically significant relationships and it consequently appears that some combination of other factors more strongly drives intraspecific size variation for most endothermic taxa.

The lack of evidence for temperature as a primary determinant of size variation in endothermic species calls into question the hypothesis that decreases in organism size may represent a third universal response to global warming. The potentially general decline in size with warming was addressed by assessments that evaluated dynamic body size responses to temperature using similar approaches to the Bergmann's rule reviews discussed above (*Sheridan and Bickford, 2011*; *Gardner et al., 2011*; *Teplitsky and Millien, 2014*). The results of these temporal reviews were similar to those for spatial relationships, but the conclusions of these studies clearly noted the variability in body size responses and the need for future data-intensive work (*Sheridan and Bickford, 2011*; *Gardner et al., 2011*) using broader temperature ranges (*Teplitsky and Millien, 2014*) to fully assess the temperature-size relationship.

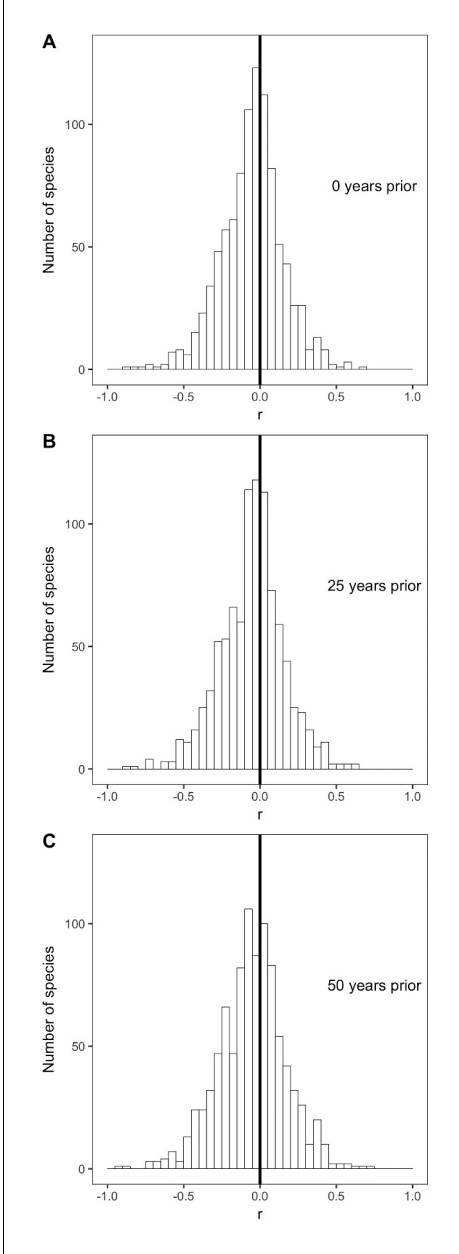

**Figure 4.** Species correlation coefficients with selected past year temperatures. Histograms of correlation coefficients (r) for all species' intraspecific temperature-mass relationships with mean annual temperature from (A) the year in which individuals were collected, (B) 25 years prior to collection year, and (C) 50 years prior to collection year. Dark vertical lines are correlation coefficients of zero. For all past year temperatures, see *Figure 4—figure supplement 1*.
DOI: https://doi.org/10.7554/eLife.27166.026
The following figure supplement is available for figure 4:

**Figure supplement 1.** Species correlation coefficients for all past year temperatures.
DOI: https://doi.org/10.7554/eLife.27166.027

Our results in combination with those from other studies suggest that much of the observed variation in size is not explained simply by temperature. While there is still potential for the size of endotherms, and other aspects of organismal physiology and morphology, to respond to both geographic gradients in temperature and climate change, these responses may not be as easily explained solely by temperature as has been suggested (*Sheridan and Bickford, 2011*; *Gardner et al., 2011*). Future attempts to explain variation in the size of individuals across space or time should use integrative approaches to include the influence of multiple factors, and their potential interactions, on organism size. This will be facilitated by analyzing spatiotemporal data similar to that used in this study, which has broad ranges of time, space, and environmental conditions for large numbers of species and individuals. This data-intensive approach provides a unique perspective on the general responses of bird and mammal species to temperature, and has potential to assist in further investigation of the complex combinations of factors that determine biogeographic patterns of endotherm size and how species respond to changes in climate.

## Materials and methods

### Data

Organismal data were obtained from VertNet, a publicly available data platform for digitized specimen records from museum collections primarily in North America, but also includes global data (*Constable et al., 2010*). Body mass is routinely measured when organisms are collected, with relatively high precision and consistent methods, by most field biologists, whose intent is to use those organisms for research and preservation in natural history collections (*Winker, 2000*; *Hoffmann et al., 2010*). These measurements are included on written labels and ledgers associated with specimens, which are digitized and provided in standard formats, e.g., Darwin Core (*Wieczorek et al., 2012*). In addition to other trait information, mass has recently been extracted and converted to a more usable form from Darwin Core formatted records published in VertNet (*Guralnick et al., 2016*). This crucial step reduces variation in how these measurements are reported by standardizing the naming conventions and harmonizing all measurement values to the same units (*Guralnick et al., 2016*). We downloaded the entire datasets for Mammalia, Aves, Amphibia, and Reptilia available in September 2016 (*Bloom et al., 2016a*, *Bloom et al., 2016b*, *Bloom et al., 2016c*, *Bloom et al., 2016d*) using the Data Retriever (*Kironde et al., 2017*; *Morris and White, 2013*) and filtered for those records that had mass measurements available. Fossil specimen records with mass measurements were removed.

We only analyzed species with at least 30 georeferenced individuals whose collection dates spanned at least 20 years and collection locations at least five degrees latitude, in order to ensure sufficient sample size and spatiotemporal extent to accurately represent each species' temperature-mass relationship. We conducted sensitivity analyses to determine if these thresholds were appropriate (*Figure 2—figure supplements 6* and *7*). We selected individual records with geographic coordinates for collection location, collection dates between 1900 and 2010, and species-level taxonomic identification, which were evaluated to ensure no issues with synonymy or clear taxon concept issues. To minimize inclusion of records of non-adult specimens, we identified the smallest mass associated with an identified adult life stage category for each species and removed all records with mass values below this minimum adult size. Results were not qualitatively different due to either additional filtering based on specimen lifestage (*Figure 2—figure supplement 4*) or removal of outliers (*Figure 2—figure supplement 5*). Temperatures were obtained from the Udel_AirT_Precip global terrestrial raster provided by NOAA from their website at http://www.esrl.noaa.gov/psd/, a 0.5 by 0.5 decimal degree grid of monthly mean temperatures from 1900 to 2010 (*Willmott and Matsuura, 2001*). For each specimen, the mean annual temperature at its collection location was extracted for the year of collection.

This resulted in a final dataset containing records for 273,901 individuals from 952 bird and mammal species (*MSB Mammal Collection (Arctos), 2015*; *Ornithology Collection Passeriformes - Royal Ontario Museum, 2015*; *MVZ Mammal Collection (Arctos), 2015*; *MVZ Bird Collection (Arctos), 2015*; *KUBI Mammalogy Collection, 2016*; *CAS Ornithology (ORN), 2015*; *DMNS Bird Collection (Arctos), 2015*; *UCLA Donald R, 2015*; *DMNS Mammal Collection (Arctos), 2015*; *UAM Mammal Collection (Arctos), 2015*; *UWBM Mammalogy Collection, 2015*; *UAM Bird*

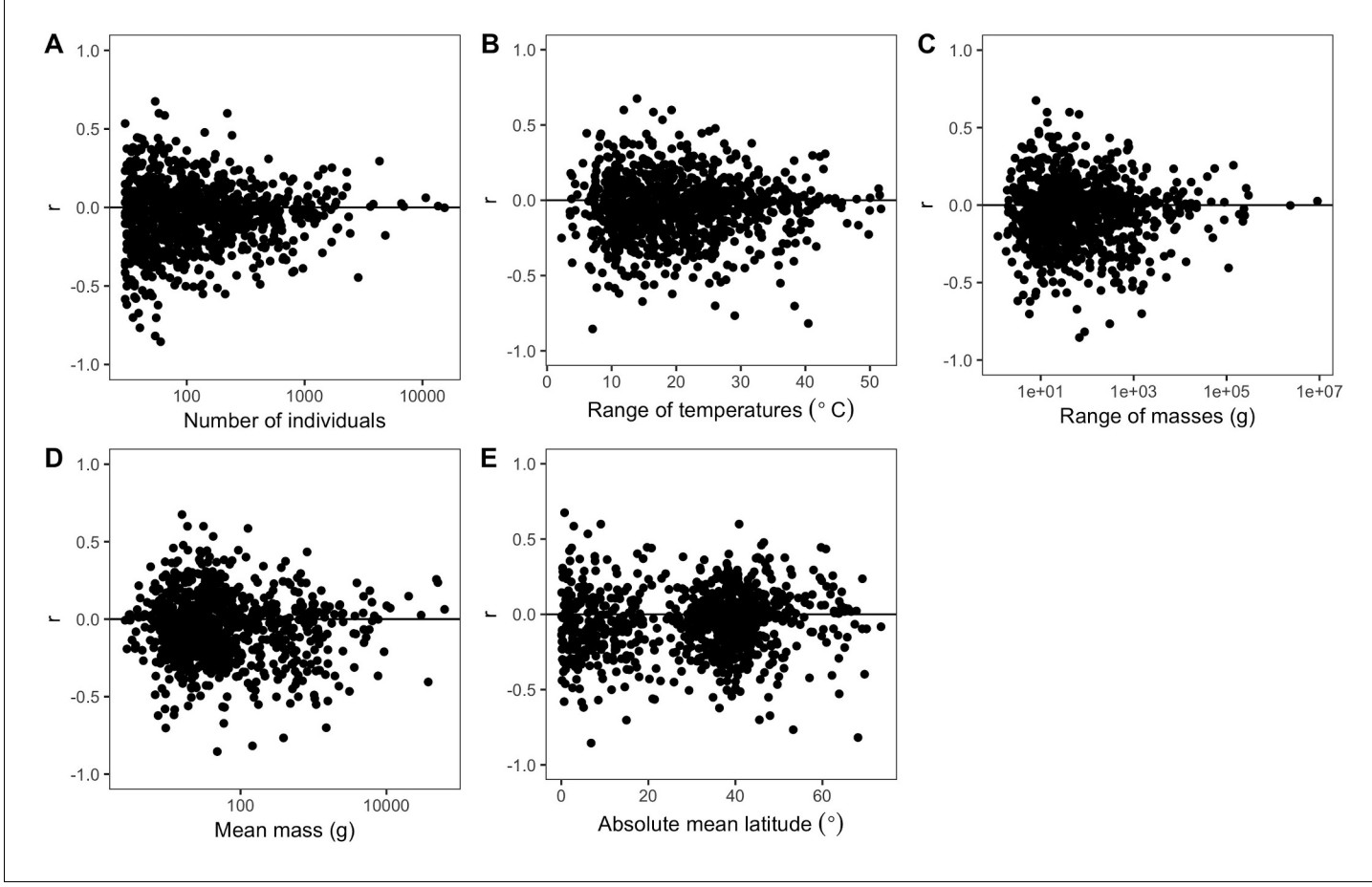

**Figure 5.** Variability of species correlation coefficients across several variables. Variation in all species' correlation coefficients (r) across the following variables for each species: (A) number of individuals, (B) difference between hottest and coldest collection year temperatures, (C) mass range, (D) mean mass, and (E) absolute mean latitude. Horizontal lines are correlation coefficients of zero. The x-axes of some plots (A, C, D) are on a log scale to better show spread of values.

DOI: https://doi.org/10.7554/eLife.27166.028

Collection (Arctos), 2015; UMMZ Birds Collection, 2015; CUMV Bird Collection (Arctos), 2015; CUMV Mammal Collection (Arctos), 2015; MLZ Bird Collection (Arctos), 2015; LACM Vertebrate Collection, 2015; CHAS Mammalogy Collection (Arctos), 2016; Ornithology Collection Non Passeriformes - Royal Ontario Museum, 2015; KUBI Ornithology Collection, 2014; MSB Bird Collection (Arctos), 2015; Biodiversity Research and Teaching Collections - TCWC Vertebrates, 2015; TTU Mammals Collection, 2015; CAS Mammalogy (MAM), 2015; Vertebrate Zoology Division - Ornithology, Yale Peabody Museum, 2015; University of Alberta Mammalogy Collection (UAMZ), 2015; UAZ Mammal Collection, 2016; Charles and Conner Museum, 2015; SBMNH Vertebrate Zoology, 2015; Cowan Tetrapod Collection - Birds, 2015; Cowan Tetrapod Collection - Mammals, 2015; NMMNH Mammal, 2015; Schmidt Museum of Natural History_Mammals, 2015; USAC Mammals Collection, 2013; MLZ Mammal Collection (Arctos), 2015; Ohio State University Tetrapod Division - Bird Collection (OSUM), 2015; Collections, 2015; DMNH Birds, 2015; CM Birds Collection, 2015; WNMU Mammal Collection (Arctos), 2015; UCM Mammals Collection, 2015; UWYMV Bird Collection (Arctos), 2015; NCSM Mammals Collection, 2015; Vertebrate Zoology Division - Mammalogy, Yale Peabody Museum, 2015; HSU Wildlife Mammals, 2016; WNMU Bird Collection (Arctos), 2015; UWBM Ornithology Collection, 2015; UCM Birds, 2015; University of Alberta Ornithology Collection (UAMZ), 2015; SDNHM Birds Collection, 2015).

The average number of individuals per species was 288, ranging from 30 to 15,415 individuals. The species in the dataset were diverse,

including volant, non-volant, placental, and marsupial mammals, and both migratory and non-migratory birds. There were species from all continents except Antarctica, though the majority of the data were concentrated in North America (*Figure 1A*). The distribution of the species' mean masses was strongly right-skewed, as expected for broad scale size distributions (*Brown and Nicoletto, 1991*), with 74% of species having average masses less than 100 g. Size ranged from very small (3.7 g desert shrew *Notiosorex crawfordi* and 2.6 g calliope hummingbird *Stellula calliope*) to very large (63 kg harbor seal *Phoca vitulina* and 5.8 kg wild turkey *Meleagris gallopavo*).

## Analysis

We fit the intraspecific relationship between mean annual temperature and mass for each species with ordinary least squares linear regression (e.g., *Figure 1B,C,D* and *Figure 1—figure supplements 1–12*) using the statsmodels.formula.api module in Python (*Seabold and Perktold, 2010*). The strength of each species' relationship was characterized by the correlation coefficient, its significance at alpha of 0.05, and the associated z score. When assessing statistical significance with large numbers of correlations it is important to consider the expected distribution of these correlations under the null model that no correlation exists for any species.

We addressed this issue by using false discovery rate control (*Benajmini and Hochberg, 1995*) implemented with the stats package in R (*R Core Team, 2016*). This method determines the expected distribution of values for p (or Z) in the case where no relationship exists for individual correlation and adjusts observed values to control for excessive false positives. Specifically, it maintains the Type I error rate (proportion of false positives) across all tests at the chosen value of alpha and therefore gives an accurate estimate of the number of significant relationships (*Benajmini and Hochberg, 1995*). This allows us to estimate the number of species with true positive and negative correlations (i.e., those that have values that exceed those expected from the null distribution). We then compared the number of species with positive and negative correlation coefficients, and the proportion of those with statistically significant adjusted p-values.

We investigated various potential correlates of the strength of Bergmann's rule. Because it has been argued that Bergmann's rule is exhibited more strongly by some groups than others (*McNab, 1971*), we examined correlation coefficient distributions within each class and order. Additionally, distributions for migrant and nonmigrant bird species were compared due to conflicting evidence about the impact of migration on temperature-mass relationships (*Ashton, 2002*). As a temporal lag in size response to temperature is likely due to individuals of a species responding to temperatures prior to their collection year (e.g., *Stacey and Fellowes, 2002*), we assessed species' temperature-mass relationships using temperatures from 1 to 110 years prior to collection year. We also examined the relationship between species' correlation coefficients and five variables to understand potential statistical and biological influences on the results. We did so with the number of individuals, temperature range, and mass range to determine if the relationship was stronger when more data points or more widely varying values were available. Because it has been argued that Bergmann's rule is stronger in larger species (*Steudel et al., 1994*) and at higher latitudes (*Freckleton et al., 2003*; *Faurby and Araújo, 2016*), we examined variability with both mean mass and mean latitude for each species. We also conducted all analyses using latitude instead of mean annual temperature. The reproducible code for these analyses is available (https://github.com/KristinaRiemer/MassResponseToTemp; *Riemer and White, 2017*) and archived (https://zenodo.org/badge/latestdoi/17957630).

## Acknowledgements

Thanks to all of the VertNet data providers, Dan McGlinn for assistance with developing this research, Rafael LaFrance for his trait extraction work, and Dave Harris for helping us divide by two.

# Additional information

## Funding

| Funder | Grant reference number | Author |
|---|---|---|
| Gordon and Betty Moore Foundation | GBMF4563 | Ethan P White |
| National Science Foundation | DEB 0953694 | Ethan P White |
| National Science Foundation | DBI 1062148 | Robert P Guralnick |

The funders had no role in study design, data collection and interpretation, or the decision to submit the work for publication.

## Author contributions

Kristina Riemer, Conceptualization, Formal analysis, Visualization, Methodology, Writing—original draft, Writing—review and editing; Robert P Guralnick, Data curation, Validation, Writing—review and editing; Ethan P White, Conceptualization, Formal analysis, Funding acquisition, Visualization, Methodology, Writing—original draft, Writing—review and editing

## Author ORCIDs

Kristina Riemer [iD] http://orcid.org/0000-0003-3802-3331
Ethan P White [iD] http://orcid.org/0000-0001-6728-7745

## Decision letter and Author response

Decision letter https://doi.org/10.7554/eLife.27166.041
Author response https://doi.org/10.7554/eLife.27166.042

# Additional files

## Supplementary files

• Transparent reporting form
DOI: https://doi.org/10.7554/eLife.27166.029

## Major datasets

The following previously published datasets were used:

| Author(s) | Year | Dataset title | Dataset URL | Database, license, and accessibility information |
|---|---|---|---|---|
| Willmott CJ, Matsuura K | 2001 | Terrestrial Air Temperature and Precipitation: Monthly and Annual Time Series (1950 - 1999) [Air Temperature Monthly Mean V3.01] | https://www.esrl.noaa.gov/psd/data/gridded/data.UDel_AirT_Precip.html | Publicly available at the NOAA Earth System Research Laboratory website (https://www.esrl.noaa.gov/) |
| Bloom D | 2016 | VertNet Amphibia | https://dx.doi.org/10.7946/P2F59W | Publicly available at Data Commons (http://datacommons.cyverse.org/) |
| Bloom D | 2016 | VertNet Aves | https://dx.doi.org/10.7946/P2K01C | Publicly available at Data Commons (http://datacommons.cyverse.org/) |
| Bloom D | 2016 | VertNet Mammalia | https://dx.doi.org/10.7946/P2TG68 | Publicly available at Data Commons (http://datacommons.cyverse.org/) |

Bloom D          2016   VertNet Reptilia          https://dx.doi.org/10.    Publicly available at
                                                  7946/P2Z59J              Data Commons
                                                                           (http://datacommons.
                                                                           cyverse.org/)

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
