## [Decision Letter]

Thank you for submitting your article "No general relationship between mass and temperature in endotherm species" for consideration by *eLife*. Your article has been favorably evaluated by Diethard Tautz (Senior Editor) and three reviewers, one of whom, Christian Rutz (Reviewer #1), is a member of our Board of Reviewing Editors. The following individual involved in review of your submission has agreed to reveal their identity: Alison Boyer (Reviewer #2).

The reviewers have discussed the reviews with one another and the Reviewing Editor has drafted this decision to help you prepare a revised submission.

Summary:

Your study uses an unprecedented dataset of animal measurements to assess the generality of one of the best-known biogeographic 'rules', and as such holds great potential for stimulating future research. The reviewers agreed that this is a noteworthy advance, but have identified a few issues that need to be addressed in a revision (essential revisions).

Essential revisions:

- Scope. You rightly note in the Introduction (first paragraph) that Bergmann's rule was originally formulated for closely-related species (i.e., interspecific patters), but you then proceed – like most earlier studies – to explore intraspecific relationships only. A comprehensive assessment of this long-standing hypothesis would cover both intra- and interspecific perspectives, and the reviewers wondered whether you could add results for the latter? These additional analyses shouldn't be too onerous, yet they would add substantial value to the manuscript. Otherwise, it should be made clear throughout that the intraspecific facet of Bergmann's rule is being examined.

- Statistical analyses. While the reviewers enjoyed the intuitive graphical illustration of results, and found that the overall patterns look compelling, they think it is essential that formal statistical analyses are conducted to support the study's conclusions. What is the null hypothesis for Figure 2 (and accompanying supporting figures)? Given the distribution of latitudes, sample sizes etc., what is the expected distribution of r values? Can Fisher's r-to-z transformation be used to calculate a null distribution, and to estimate the excess of positive and negative values?

- Migratory birds. The results for birds are complicated because many of them are migratory (e.g., in the USA, birds migrate between North and South America). This is an important confounding factor, as for almost all species, it is impossible to satisfactorily define the latitude at which they exist (in fact, this problem may even apply at smaller scales, if species move in response to extrinsic factors, such as harsh weather conditions). This alone argues against the generality of Bergmann's rule, and suggests that it has little applicability to birds. Please formally explore the effect of this confound on the overall patterns observed, for example, by re-running analyses with (migratory) birds excluded.

[Editors' note: further revisions were requested prior to acceptance, as described below.]

Thank you for sending your article entitled "No general relationship between mass and temperature in endothermic species" for peer review at *eLife*. Your article is being evaluated by one peer reviewer, and the evaluation is being overseen by a Reviewing Editor and Diethard Tautz as the Senior Editor.

One of the original reviewers has kindly provided further comments on your statistical analyses, and we would appreciate if you could briefly respond to these, as there may have been a misunderstanding:

They clarified that their point was that you could have more simply converted the r -> z and overlain a z distribution rather than highlighting statistically significant values. They noted that overlaying a z-distribution on z-transformed r values would show the extent of disagreement or not in the tails more clearly. They also commented that r is just bounded at -1 and 1, so the information is quite limited, and that overall, r is not a particularly informative comparative measure of effect size for heterogeneous data varying in sample size etc.

---

## [Author Response]

Essential revisions:- Scope. You rightly note in the Introduction (first paragraph) that Bergmann's rule was originally formulated for closely-related species (i.e., interspecific patters), but you then proceed – like most earlier studies – to explore intraspecific relationships only. A comprehensive assessment of this long-standing hypothesis would cover both intra- and interspecific perspectives, and the reviewers wondered whether you could add results for the latter? These additional analyses shouldn't be too onerous, yet they would add substantial value to the manuscript. Otherwise, it should be made clear throughout that the intraspecific facet of Bergmann's rule is being examined.

We have decided, after extensive discussion, that adding interspecific analyses of Bergmann’s rule would make the current manuscript too complex and difficult to follow and therefore these analyses warrant their own manuscript. The reason for this is that the interspecific analyses in the literature have been conducted in a number of different ways, none of which immediately aligns with large scale individual-level data. The most common older form of this analysis involves assessing changes in the average size of a species as a function of the latitudinal centroid of its geographic distribution, where each point is a species within a genus. Two examples from the literature illustrate the variety in this type of analysis: correlations between body length and mean breeding range latitude for species within genus, family, and order (Boyer et al., 2010) and correlation between geometric mean mass and six environmental variables, including a phylogenetic component, for species within an order (Diniz-Filho et al., 2007). Newer analyses do something similar using range maps for entire assemblages and look at how the average size of all species in the assemblage varies across grid cells in response to environmental factors, as in Blackburn and Gaston (1996). Neither of these approaches are well suited to analysis using VertNet data because these data are not necessarily collected broadly or evenly across each species’ geographic range.

This diversity of approaches comes from the range of conclusions that have been reached about the contents of Bergmann (1847). Bergmann states that the pattern should apply across “races”. This is difficult to interpret, especially given how taxonomic classification has changed. It is generally agreed upon that “closely related species” is analogous, though this is only somewhat less vague in terms of selecting useful and appropriate analyses. Many of the interspecific approaches taken also have statistical and inference limits that need to be explored. Consequently, addressing the pattern more broadly is quite a bit more complicated than, e.g., using genus instead of species to group individual-level sizes. As a result of these complexities, a sufficient exploration of interspecific Bergmann’s rule would require additional data, new analyses, and the space of a full manuscript to explain and explore this approach.

That said, we certainly agree that this is an important topic to explore, including in new ways using the kinds of data used in this paper. Therefore, we have added a paragraph to the Discussion summarizing the importance and challenges of pursuing this question as in future research. We have also added language throughout the manuscript to emphasize that the analyses pertain to the intraspecific version of Bergmann’s rule.

- Statistical analyses. While the reviewers enjoyed the intuitive graphical illustration of results, and found that the overall patterns look compelling, they think it is essential that formal statistical analyses are conducted to support the study's conclusions. What is the null hypothesis for Figure 2 (and accompanying supporting figures)? Given the distribution of latitudes, sample sizes etc., what is the expected distribution of r values? Can Fisher's r-to-z transformation be used to calculate a null distribution, and to estimate the excess of positive and negative values?

If we understand correctly, we believe that this question reflects a failure on our part to clearly communicate the analyses that we have already conducted. The null hypothesis for Figure 2 is that no species has an intraspecific relationship between temperature and mass. Given the distribution of temperatures/latitudes and sample sizes, this null hypothesis would lead to a distribution of correlation coefficients roughly centered on zero with some species showing larger positive and negative values of r by chance and some of these relationships (roughly 5%) appearing to be statistically significant at p < 0.05 based on their Z scores. The standard approach to assessing the expected (null) form of this distribution and to estimate the number of excess positive and negative values is by controlling the false discovery rate (Verhoeven et al., 2005; Garcia, 2004; Pike, 2010; Waite and Campbell, 2006; Nakagawa, 2004). This analysis has already been conducted and presented in Figure 2. In that figure, r values falling within the null distribution are presented in white and the “excess” negative and positive values are shown in purple and green, respectively. As is standard when controlling the false discovery rate, we did the calculations on the p-values which are calculated from Z scores. Therefore it is our understanding that we have already performed the requested analysis. If we have misunderstood, we would be happy to conduct additional analyses.

We clearly failed to discuss this analysis in sufficient detail to communicate effectively. Consequently we expanded our description of how we assess r and p-values in the Materials and methods, including their expected distributions under both the null and alternative hypotheses. Additionally, we better explained the role of false discovery rate control and what it accomplishes in identifying those species with excess positive and negative relationships beyond the null, and the proportion of species that have no relationship between temperature and mass.

García, Luis V. “Escaping the Bonferroni Iron Claw in Ecological Studies.” Oikos 105, no. 3 (2004): 657–63. doi:10.1111/j.0030-1299.2004.13046.x.

Nakagawa, Shinichi. “A Farewell to Bonferroni: The Problems of Low Statistical Power and Publication Bias.” Behavioral Ecology 15, no. 6 (2004): 1044–45. doi:10.1093/beheco/arh107.

Pike, Nathan. “Using False Discovery Rates for Multiple Comparisons in Ecology and Evolution.” Methods in Ecology and Evolution 2, no. 3 (2011): 278–82. doi:10.1111/j.2041-210X.2010.00061.x.

Verhoeven, KJF, KL Simonsen, LM Mcintyre, Source Oikos, and Fasc Mar. “Implementing False Discovery Rate Control : Increasing Your Power False Discovery Rate Control : Implementing Increasing Your Power.” Oikos 108, no. September 2004 (2005): 643–47.

Waite, Thomas A., and Lesley G. Campbell. “Controlling the False Discovery Rate and Increasing Statistical Power in Ecological Studies 1.” Ecoscience 13, no. 4 (2006): 439–42.

- Migratory birds. The results for birds are complicated because many of them are migratory (e.g., in the USA, birds migrate between North and South America). This is an important confounding factor, as for almost all species, it is impossible to satisfactorily define the latitude at which they exist (in fact, this problem may even apply at smaller scales, if species move in response to extrinsic factors, such as harsh weather conditions). This alone argues against the generality of Bergmann's rule, and suggests that it has little applicability to birds. Please formally explore the effect of this confound on the overall patterns observed, for example, by re-running analyses with (migratory) birds excluded.

This is a really important point, and we thank the reviewers for catching it. We have added separate analyses on bird species in our dataset known as migrants or nonmigrants, see new Figure 2—figure supplement 1. The proportion of species with negative statistically significant, positive statistically significant, and no relationships were similar (i.e., varying by no more than one percentage point) between the migrant species and nonmigrant species. The mean correlation coefficients for migrant species and nonmigrant species were -0.06 and -0.07, respectively. There is a somewhat more apparent shoulder of small r values below zero, but these are all within the null distribution. Therefore our assessment of the new results is that migration had minimal impact on the conclusions of this manuscript. We have added the appropriate text for this figure to the Materials and methods and Results sections.

[Editors' note: further revisions were requested prior to acceptance, as described below.]

One of the original reviewers has kindly provided further comments on your statistical analyses, and we would appreciate if you could briefly respond to these, as there may have been a misunderstanding:They clarified that their point was that you could have more simply converted the r -> z and overlain a z distribution rather than highlighting statistically significant values. They noted that overlaying a z-distribution on z-transformed r values would show the extent of disagreement or not in the tails more clearly. They also commented that r is just bounded at -1 and 1, so the information is quite limited, and that overall, r is not a particularly informative comparative measure of effect size for heterogeneous data varying in sample size etc.

We received a very thoughtful comment from one reviewer about the benefits of the inclusion of the z scores for each species in our dataset. Consequently, we have included a figure of these z scores in the supplemental material for this manuscript, with corresponding edits to the manuscript text.